# Epidemiological, serological, and viral genomic analysis of an outbreak of elephant hemorrhagic disease in Switzerland

**Mathias Ackermann**[1]*, **Jakub Kubacki**[1,2,3], **Sarah Heaggans**[4], **Gary S. Hayward**[4], **Julia Lechmann**[1]

**1** Institute of Virology, Vetsuisse Faculty, University of Zurich, Zurich, Switzerland, **2** Institute of Virology and Immunology, Mittelhäusern, Switzerland, **3** Department of Infectious Diseases and Pathobiology (DIP), Vetsuisse Faculty, University of Bern, Bern, Switzerland, **4** Viral Oncology Program, Johns Hopkins School of Medicine, Baltimore, Maryland, United States of America,

* mathias.ackermann@uzh.ch

## Abstract

Elephant hemorrhagic disease (EHD), caused by several Elephant endotheliotropic herpesviruses (EEHV), represents a frequently lethal syndrome, affecting both captive and free-living elephants. In the summer of 2022, three young Asian elephants (*Elephas maximus*) succumbed to EHD in a zoo in Switzerland, despite considerable preventive efforts and early detection of EEHV1A viremia. In this communication, we describe the extent of preventive measures in terms of prior virus detection, active survey of viremia, and antibody status. The results show that: (1) A previously undetected EEHV1A strain had remained unrecognized among these elephants. Probably, the virus re-emerged after almost 40 years of latency from one of the oldest elephants in the zoo. (2) While two of the three affected animals had prior immune responses against EEHV1, their strain-specific immunity proved insufficient to prevent EHD. The complete genomic DNA sequence of the EEHV1A strain involved was determined, and detailed comparisons with multiple EEHV1 strains were made, revealing a much greater extent of divergence and level of complexity among the encoded proteins than previously described. Overall, these data confirmed that all three EHD cases here had been infected by the same novel strain of EEHV subtype 1A.

## Introduction

Elephant hemorrhagic disease (EHD) can be caused by several Elephant endotheliotropic herpesviruses (EEHV, members of the genus *Proboscivirus* within the family of *Orthoherpesviridae*) and represents a frequently lethal syndrome, affecting both captive and free-living elephants [1–4]. Seven distinct but genetically related species of EEHV, which fall into both an AT-rich and GC-rich branch, are known to infect either Asian (*Elephas maximus*) or African elephants (*Loxodonta africana*) [5,6]. EEHV type 1 (EEHV1), which has been most frequently associated with the occurrence of EHD in Asian elephants, can be further divided into the chimeric subtypes A (EEHV1A) and B (EEHV1B), as well as into cladal ORF-Q protein subtype clusters encoded by gene E39 [4,6,7]. The knowledge of the circulating viral subtypes and strains is important for prognosis and treatment: subtype 1A is considered the

**Data availability statement:** All raw sequencing data generated in this study have been deposited in the Sequence Read Archive (SRA) under BioProject ID: PRJNA1052582 (https://www.ncbi.nlm.nih.gov/bioproject/PRJNA1052582/). The full sequence of EP55 has been submitted to GenBank and is available under accession number OR543011 (https://www.ncbi.nlm.nih.gov/nuccore/OR543011).

**Funding:** This work was supported by a private donation from the late Prof. Dr. Robert Wyler to M.A. (D-52601-01-01) and by general funds allocated to J.L. by the Institute of Virology, University of Zurich, Switzerland. G.S.H. and S.Y.H. received financial support from the International Elephant Foundation (IEF), a non-profit organization that facilitates donations for elephant conservation research projects worldwide. While G.S.H. is currently a member of the IEF Advisory Board, the funders had no role in the study design, data collection and analysis, decision to publish, or preparation of this manuscript. However, IEF has previously assisted Johns Hopkins University investigators by coordinating the collection and transport of necropsy and other pathological samples for PCR strain and subtype DNA sequencing analysis.

**Competing interests:** The authors have declared that no competing interests exist.

most virulent EEHV, whereas subtype 1B is known to frequently cause milder courses of EHD [1,8].

In African elephants, EEHV6 is the genetic counterpart of EEHV1 [9]. Among others, EEHV6 has been linked with several cases of EHD in African elephants in both Europe and the USA [1]. The other five EEHV species are represented by EEHV2, EEHV3 and EEHV7 in African elephants [4,10,11] plus EEHV4 and EEHV5 in Asian elephants [1,6,11].

As with all herpesviruses, the reservoir of EEHV is in latently infected animals, which may intermittently reactivate and shed virus for transmission to herd mates [1,12]. While all known EEHV species replicate in the respiratory tract for virus excretion and transmission, viremia may also occur intermittently, which is the major precondition for developing clinical EHD [13]. Another critical parameter for developing EHD is the level of specific anti-EEHV antibodies in the serum. Very young animals are thought to be protected by maternal antibodies. During this period, subclinical infections may take place, which induces active immunological protection against the disease later in life [7,14,15]. Accordingly, the highest risk for young animals to succumb to EHD is during and right after weaning (2-8y), in association with the waning of maternal antibodies [7]. However, the study of epidemiology, pathogenesis, treatment, and prevention of EEHV-related diseases has been hindered because, as yet, none of the EEHV species or strains can be propagated in conventional cell cultures. Moreover, animal models for EEHV and EHD do not exist, because no other susceptible species have been identified.

Recent partial and full genomic DNA sequence analysis of many EEHV species and strains have revealed large genetic differences between them [4,9–11,16]. Both EEHV1A and EEHV1B genomes total 180-kbp in size and comprise approximately 120 genes [5,17] of which two-thirds do not have orthologues in other herpesviruses. To compare entire EEHV genomes, they can be envisaged as being divided into seven segments, i.e., C1 and C2, harboring the core conserved genes, as well as L1, L2, L3, R1, and R2 segments, which encompass genes that are unique to the probosciviruses [4,5]. Numerous other sequence variations are scattered all over the genome, but four major non-adjacent and highly diverged loci (CD-I, CD-II, CD-III as well as R2) have been described [4,5]. Notably, several additional hypervariable loci with multiple subtypes mapping across other parts of both the EEHV1A and EEHV1B genomes are neither linked to each other nor to the classical A versus B diaspora [4,16,17] (for more details see S1 File).

The multitude of these differences and variations may be highly relevant to EHD, since even most of the likely envelope glycoproteins are highly divergent, including gB, gH, gL, and the three adjacent spliced glycoproteins ORF-O, ORF-P and ORF-Q that are expected to play major roles in immunity and serology [7,14]. Lack of antibodies against the same ORF-Q antigen clade as that of the infecting virus strain have been associated with vulnerability to death from EHD [7]. However, that same variability complicates discovering interconnected cases of EHD by means of just partial DNA sequencing.

Only recently, the value of serology has also been established as a useful tool in the context of EEHV epidemiology as well as in EHD risk management [2,7,14,15].

Albeit highly diverged, all known species of EEHV encode thymidine kinase (Tk) as well as conserved protein kinase enzymes (CPK), i.e., U48.5/EE7 encoding the EEHV version of Tk (ETk) and U69 encoding the EEHV version of CPK (ECPK), respectively. It has been speculated that these viral enzymes may be capable of phosphorylating nucleoside analogues such as ganciclovir (GCV), aciclovir (ACV), and penciclovir (PCV) or famciclovir (FCV, the PCV-derivate for oral application), which is essential for the potential activity against EEHV of these antiviral drugs [16–20]. Indeed, nucleoside analogues such as ACV, FCV, and GCV

have been used to treat elephants with clinical and virological EHD, but the clinical efficacy of these drugs remains debatable [1,21–23], although intravenous treatment with GCV coincided at least in one case with decreasing viremia [24]. However, in the absence of animal models and suitable cell cultures, the true value of treatment against EEHV-infections with nucleoside analogues remains unknown [1].

Despite all these uncertainties, the desire to prevent lethal cases of EHD among young elephants in captivity increased with the growing knowledge about its causative agents and their biological properties. Accordingly, many zoos initiated surveillance programs to identify prevalent EEHV types and strains, and to monitor their animals at risk for early signs of viremia [13,25]. It was expected that typing of the prevalent EEHV species and strains may help to focus on particularly dangerous isolates, while screening for viremia may allow early onset of antiviral treatment in cases of emergency [8,21,24,26].

Prior to 2022, three cases of lethal EHD had been recorded among Asian elephants in Switzerland, all of them attributable to EEHV1A, the first in 1988 (partial sequences deposited in Genbank as EP04, Lohimi: KT705308.1), the second in 1999 (partial sequences in GenBank as EP07, Xian: MH287538.1) and the third in 2003 (EP16, Aishu; no sequence entries in GenBank but unpublished U48.5/EE7 sequences, encoding for ETk, classified the isolate as subtype 1A) [25,27]. In 2014, after many years of successful breeding in a closed collection, a 10-year-old male (Thai, see Materials and Methods and Fig 1) was introduced to the herd, with the aim to replace the aging prior breeding bull (Maxi) and to avoid inbreeding. As a calf had been born in the same year (Omysha), a protocol was established, which included risk assessment and active surveillance. Initially, EEHV-shedding was assessed, which resulted in the detection of two additional circulating viruses among these elephants, namely an EEHV1B isolate from Omysha as well as an EEHV4 isolate from Thai [25]. In 2017 and 2020, respectively, two additional elephant calves were born in the zoo, both of which had been fathered by the newly acquired breeding bull. Starting at the age of one year, the young elephants were

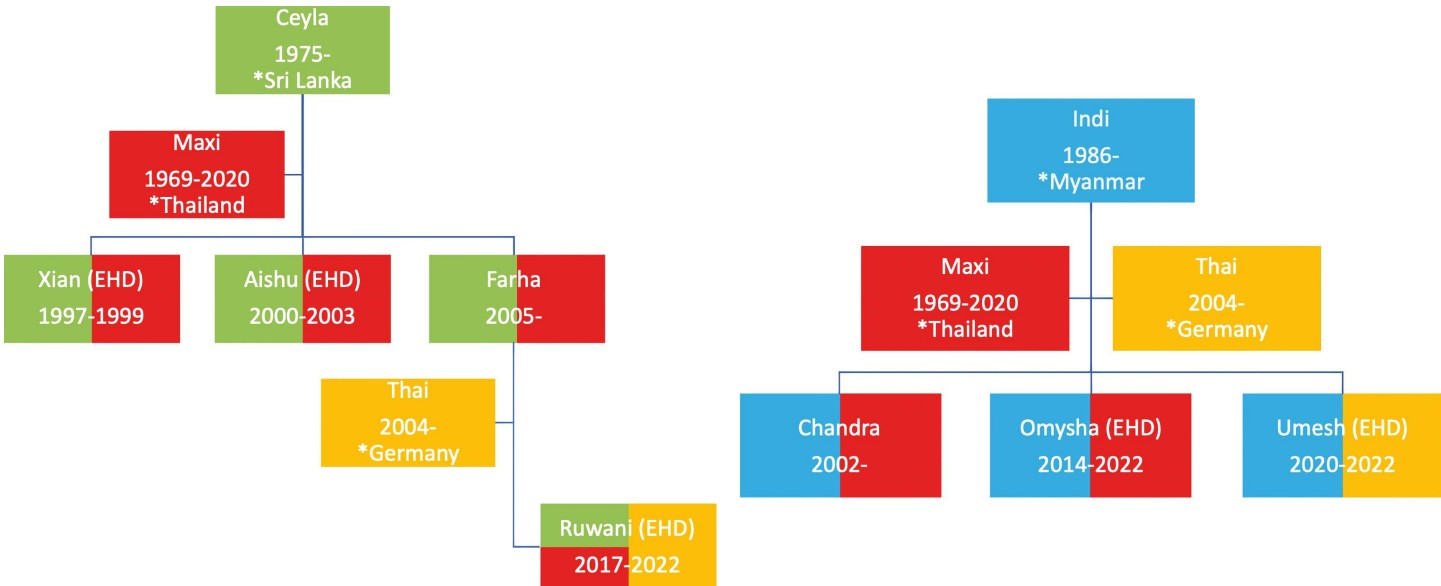

**Fig 1. Family trees of the elephants relevant for the present study.** Green and blue: founder females with year and country of birth. Red and orange: breeding bulls. Multicolored: color coded offspring with year of birth and year of death (if applicable). Two prior cases of EHD (Xian, Aishu) are also integrated. EHD in brackets: animal was lost to EHD.

considered as being at risk to succumbing to EHD. Accordingly, they were included into the preventive program and screened on a weekly basis for viremia by EEHV1 and EEHV4 [26]. Serology was established between 2020 and 2022, as it became available from literature [2,7,14]. Moreover, stocks of GCV and FCV were acquired for treatment of clinical EHD cases.

In summer 2022, the three juvenile elephants from this herd died of EHD caused by a previously undetected EEHV1A strain. This paper describes our attempts at identification of the source of this outbreak through genomic DNA sequencing of the causative virus as well as to evaluate its relationship to other well-characterized EEHV1 strains and for identifying encoded antigenic proteins relevant for developing specific serological assays.

Our analysis of these new cases included a full evaluation of the single novel EEHV1 (Umesh, EP55) strain involved and its patterns of divergence at both the intact genomic and single protein-coding gene levels from other previously encountered or published EEHV1 reference strains.

## Materials and methods

### Ethical statement

Viremia samples, tissue samples and sera from elephants were analyzed by order of the contributing zoos. Ethical approval was not sought as the present study only made use of samples that were taken for specific diagnostic purposes, the results of which are described in this manuscript. The samples were taken for routine veterinary care by veterinary staff. The contributing zoos as well as their scientific advisors consider it important to sharing critical medical, husbandry, training, and scientific information with colleagues across the globe.

### Elephants

At the time of the outbreak, the herd of Asian elephants (*Elephas maximus*) consisted of eight animals, divided into two family lines and one breeding bull, who had been introduced from Germany in 2014. Also in 2014, the management of the elephants was changed from "hands on" to "protected contact" principles, which implied that all elephant-human interactions, including all methods of sampling, were determined by the individual elephant's decision to interact [28]. The two family lines lived separately but were able to directly interact with each other. Fig 1 gives an overview of the relationships between the individual elephants as well as their life span and origin, if they were born outside of the Zurich zoo. In addition to the present animals, three additional males are indicated in the figure, one (Maxi) being the former breeding bull and the other two representing two previous cases of EHD (Xian, EP07 and Aishu, EP16, respectively) in the same zoo.

The first family line, whose founder Ceyla had been born in 1975 in Sri Lanka and transferred to the zoo in 1976, consisted of three individuals, including the founder's daughter Farha and her daughter (Ruwani), born to Farha and Thai, the new breeding bull. Notably, Xian and Aishu had also been members of this family line but had been fathered by the previous breeding bull Maxi.

The second family line, whose founder Indi had been born in 1988 in Myanmar and transferred to the zoo in 1999, consisted of four individuals, including the founder's two daughters from Maxi (Chandra and Omysha) and her recent son (Umesh) from Thai.

Thus, all individuals except of the two founders and the bulls had been born in the Zurich zoo.

## Samples

**Viremia samples.** For weekly viremia screens, one blood droplet (approx. 20 μL) was drawn from the elephant's feet during pedicure by using a small capillary, which then was placed into an Eppendorf tube containing 80 μL nuclease-free water and thoroughly mixed. Due to the "protected contact" principles, the animals chose to occasionally refuse the sampling procedure. Omysha's sampling period started on January 21, 2018, whereas sampling of Ruwani and Umesh started on July 3, 2018 and February 3, 2021, respectively. Routine screening ended with the first detection of serious EEHV1 viremia in Umesh's sample on June 23, 2022.

Historic EDTA blood samples from EHD cases in 2018 (H1 Anjuli and H2 Kanja) were kindly provided by the Hagenbeck zoo in Hamburg, Germany. Cellular fractions were taken as source of DNA extraction for PCR, whereas plasma fractions were used in serology.

**Serum samples.** Sera from Swiss elephants are routinely collected and archived for health monitoring purposes. Samples from these archives were provided for serological analyses by order of the Basel zoo and in the Zurich zoo, respectively. Samples from the Zurich zoo were collected throughout March 2022. Four sera (A1, A2, A3, A4) from African elephants in Basel zoo were collected throughout January 2022.

**Tissue samples.** After necropsy of each of the three elephants, 15 mg from each of five tissues (heart, spleen, liver, kidney, lung) were combined from each individual and minced with 300 μL of body fluid and 300 μL blood using a TissueLyser II (Qiagen).

In addition, a few slices of historic heart and liver tissues from Xian, son to Maxi and Ceyla, who had been lost to EHD in 1999 (partially sequenced genome in GenBank as Xian, EP07), were made available for the present study.

## Real-time PCR detection of EEHV1, EEHV4, and ETNF

The previously described real-time PCRs for EEHV1 (targeting U41, the major DNA-binding protein locus) and EEHV4 (targeting U60, the terminase subunit 1 locus), respectively, identifying all known subtypes of a particular virus type [13,25]. The elephant TNF-alpha (ETNF) locus was additionally targeted for two purposes, (1) to test for amplifiable elephant DNA in the sample and (2) as a means to quantify the relative amount of viral DNA in the sample compared to host DNA [13]. Briefly, 5 μL aliquots from diluted blood droplet samples were used as template per well. In addition, each 20 μL reaction volume contained 10 μL PrimeTime Gene Expression Master Mix (Integrated DNA Technologies), 0.9 μM of each primer and 200 nM probe, and 5 μL template or nuclease-free water as negative control. Samples were tested in duplicate on a QuantStudio 3 Real-Time PCR System (Applied Biosystems) using the following cycling conditions: 2 min at 50°C, 10 min at 95°C, 40 cycles of 15 sec at 95°C and 1 min at 60°C.

A test was considered valid if ETNF was detected at a Ct value of 39 or lower and if the negative controls for each assay did not yield a Ct value. To compensate for sample variation in volume and elephant cells, the Ct value of ETNF was used as an approximate measure for the number of elephant cells within the sample. As the virus target is present at one copy per virus genome, and the host target is present at two copies per diploid genome, the ETNF value is proportional to the cell number in the assay and, thus, can be used to measure the proportion of viral genomes and cellular genomes. Therefore, the Ct value of the ETNF signal was subtracted from the Ct value for EEHV to get a ΔCt value ($\Delta Ct = Ct_{EEHV} - Ct_{ETNF}$). Accordingly, the ΔCt values grew on the negative side with increasing numbers of EEHV genomes per cell.

## Timeframe of the outbreak

The outbreak was first detected on June 23rd, 2022, with the first elephant (Umesh, see Fig 1) becoming terminally viremic (day 1), and ended with the death of Ruwani on July 22nd (day 30).

**Conventional PCRs.** In a first round, a previously established PCR targeting the E36/U79 locus to discriminate between EEHV1A and EEHV1B strains was used [25]. In a second round, the E54/EE51 vOX2–1 locus was targeted as it is stable in separate isolates from the same source, while isolates from differing sources may differ from each other by diverging up to 15% at both the nucleotide and the aa level [29]. The oligonucleotides used in these PCRs are listed in Table 1. The PCRs were run in a volume of 20 μL, comprising 5 μL template, 10 nM dNTP, 5x Phusion GC buffer, 2 U of Phusion High-Fidelity DNA Polymerase (Thermo Scientific) and each primer at a concentration of 0.5 μM (Table 1). After an initial denaturation for 2 min at 98°C, the samples were cycled 39 times at 98°C for 10 sec, 58.4°C for 20 sec, and 72°C for 7 sec. The runs were completed with 10 min at 72°C before cooling to 10°C. The PCR products were evaluated for their size by agarose gel electrophoresis before being extracted from the gel for cycle sequencing by a commercial company (Microsynth, Balgach, Switzerland).

## LIPS assays

**LIPS antigens.** The selection and production of LIPS antigens was modified from a previously published protocol [7]. Antigen for LIPS serology was harvested from transfected COS7 cells, using synthetic and codon-optimized DNA (GenScript Europe, the Netherlands). For this purpose, the published amino acid sequences of E39 clades A (Kimba), B (Emelia), C (Raman), and D (Daizy) encoding ORF-Q [7] as well as of the gB glycoprotein (EEHV1A, Kimba, [18]) were reverse translated, codon-optimized for expression in human cells, and cloned under the control of the cmvIE promoter into the pcDNA3.1(+) vector. Moreover, each construct was fused with the following sequence of C-terminal tags: V5-epitope, nanoLuciferase (nLuc), and 7-his. Except for ORF-Q clade D, each construct possessed its native signal peptide, which was included into the gene synthesis design. As clade D was not predicted to possess a functional signal sequence but two short transmembrane regions close to the N-terminus, the molecule was shortened to start only at its original amino acid No. 60, which was preceded by the clade A signal peptide. Expression of the desired antigens following transfection was verified by measuring nLuc activity (relative light units, RLU) in the cell culture supernatants as well as by detecting the corresponding proteins in cell lysates after standard Western immunoblotting using a commercial anti-V5 monoclonal antibody (SV5-Pk1 from Invitrogen).

**LIPS assay.** The protocol for the LIPS assay was adapted from a previously published overview article [30]. A commercial kit (Immunoprecipitation Kit, Abcam, Cambridge, UK), which provided buffers for protein extraction and solubilization as well as Protein A/G sepharose beads and washing buffers, was used according to the supplier's instructions. Briefly, the secreted antigens were harvested from the cell culture supernatants and mixed

**Table 1. Oligonucleotides used for conventional PCR.**

| Target | Oligo | 5' to 3' sequences | Amplicon size |
|---|---|---|---|
| E36/U79 | F-E36 | TCC AGG GAT TTC TCC AGT TG | 126 bp |
| | R-E36 | GCC ACC TTC TTC TGC TTT TG | |
| E54/EE51 | AttB1_vOX_R1 | GGGGacaagtttgtacaaaaaagcaggctATGCT[T/**C**]CAGAGAAAGTACAGGTAC[a] | 930 bp |
| | AttB2_vOX_L1 | GGGGaccactttgtacaagaaagctgggtGTGTTGCCG[T/**C**]CACGATGC**C**TTCTACG[b] | |

The primers targeting the vOX2–1 locus, encoded by E54/EE51, were elongated by att-recombination sequences (lowercase letters) for facilitated cloning and sequencing after PCR amplification. Moreover, an equimolar mixture of two different forward (R1 with wobbling at position 6)[a] and reverse (L1 with wobbling at position 10)[b] primers was used to compensate for published sequence variations at this site.

with non-denaturing solubilization buffer (NDSB) and protease inhibitors before non-soluble proteins were removed by centrifugation. Prior to the assay, the nLuc activity of each antigen was adjusted to comprise $10^7$ RLU per second (RLU/s) in a volume of 50 μL (NanoGlow, Promega, Madison, Wisconsin, USA). The sera were prediluted with phosphate buffered saline (PBS) to 1:100 in a polystyrene microtiter plate (Nunc F96 MicroWell flat bottom) and mixed well by shaking. These were kept in these plates at -20°C for a maximum of three weeks. On a second polystyrene plate, 40 μL of NDSB were pipetted into each well before 10 μL of prediluted sera and 50 μL of freshly diluted nLuc-antigen were added. The mixture was incubated by shaking (300 rpm) for one hour at ambient temperature. Meanwhile, a slurry comprising 30% sepharose A/G beads was prepared in PBS and 5 μL were added to each serum sample to incubate for one more hour with shaking. Next, all samples were transferred to a filter plate (MultiScreen, Merck, Zug, Switzerland) and washed on a vacuum manifold. Each well was washed eight times with 100 μL of wash buffer, followed by two times washing with PBS. Following the last wash, the vacuum was turned off. The filter plate was removed and blotted dry using a stack of filter paper making sure to remove moisture on the top and bottom of the plate. Subsequently, 50 μL of NanoGlow substrate (Promega) was added to each well before measuring in a luminometer (MicroLumat Plus, LB 96 V, up-version 2.0; software WinGlow v. 1.25.000003, Berthold Technologies, Bad Wilbad, DE). Since horses are expected to be free of antibodies against any EEHV, two horse sera (P1 and P2), both known to have antibodies against equine herpesvirus 1 (EHV-1) were used as controls to determine the negative cut-off values of each test. As no true EEHV-negative elephant sera are available, the cut-off decisions were based on a previous publication on marine turtle serology [31].

## Sample Preparation and DNA Sequencing

DNA was extracted from necropsy samples for whole genome sequencing. A previously established viral particle enrichment protocol was used to increase the recovery of viral genetic material [32]. Enrichment of viral nucleic acids was followed by reverse transcription for any RNA viruses (to exclude other causes of death) and sequence-independent single primer amplification for DNA. The nucleic acids were sheared to 500 bp length using the E220 Focused-ultrasonicator (Covaris, Woburn, MA, USA) and prepared with the NEBNext Ultra II DNA Library Prep Kit for Illumina (New England Biolabs, Ipswich, MA, USA) according to the manual. Sequencing libraries were sequenced at the Functional Genomics Center Zurich (FGCZ) in a paired-end 2 × 150 bp, SP flow cell sequencing run using the NovaSeq 6000 (Illumina, San Diego, CA, USA). The PhiX Control v3 Library (Illumina, San Diego, CA, USA) was added as the control.

## Data analysis

Sequenced reads were analyzed using the *de novo* assembly pipeline described previously [33]. Illumina sequencing adapters and low-quality sequencing ends were trimmed using Trimmomatic (v. 0.39) [34]. Subsequently, trimmed reads with average quality above 20 and longer than 40 nt were further cleaned up by trimming away the SISPA primers using cutadapt (v. 3.2). The quality-checked reads were then assembled using metaspades (v. 3.12.0) [35]. Assembled *de novo* contigs were compared to the National Center for Biotechnology Information (NCBI) non-redundant database (nt) using BLAST-N (v. 2.10.1+) and annotated using the best BLAST-N hits. Quality-checked reads were mapped back to the assembled sequences using bwa (v0.7.17) mem [36]. In addition, contigs from EEHVs were aligned in a metagenomic pipeline of the SeqMan NGen v.17 (DNASTAR, Lasergene, Madison, WI, USA) to all available viral genomes of other EEHVs downloaded from the NCBI database (08.2022) and visualized in the SeqMan Ultra (DNASTAR, Lasergene, Madison, WI, USA).

To determine general similarity values between our assembled Umesh strain and several other prototype fully sequenced EEHV genomes, including two EEHV1A strains, as well as EEHV1B and EEHV6, we initially performed a variety of standard NCBI BLAST-N and BLAST-P searches. Later, after adding seven more intact genomes from GenBank, including six from India [3,29], they were also subjected to more extensive comparative analysis using the advanced VIRIDIC version of BLAST-N (settings: '-word_size 7 -reward 2 -penalty -3 -gapopen 5 -gapextend 2') [37].

## Annotation of EEHV genomes

Typical EEHV genomes encompass about 40 genes that are also found as conserved but characteristically diverged versions within the majority of other mammalian herpesvirus sub-families, and are referred to as "core" genes with a U number gene nomenclature system derived from that used originally for human HHV6 genome analysis [38]. The remaining 80 or so *Probocivirus*-specific protein encoding genes that have not been found in any other herpesviruses have been given a sequential unique E number nomenclature system based on that originally applied to the prototype EEHV1A (Kimba) genome [4,5,13]. Universally agreed upon three-letter abbreviations (e.g., POL) were used to name proteins with previously available functional or homology data from other well-studied herpesvirus species. Otherwise, we used a combination of either an extended ORF-A to ORF-S system as originally suggested by Ehlers et al [17] or just left them with the same E number as for the designated gene name where little else is known for a novel *Probocivirus*-specific protein. In terms of detailed gene-by-gene analyses, there are two alternative published physical genetic maps of prototype EEHV1A genomes that Umesh can be compared to with that of EEHV1A (Raman) from Wilkie et al [16] being drawn in the opposite orientation and using a different (EE number) gene nomenclature system. Furthermore, the Umesh genome described here and most other intact annotated AT-rich branch genomes of EEHV1, EEHV2A, EEHV5B and EEHV6 strains as well as those of the GC-rich branch genomes EEHV3A, EEHV3B and EEHV4B in GenBank have all been given the same orientation and E number nomenclature as that used for Kimba [18], except for having their R2-segments switched to the extreme left-hand-side compared to our original inadvertent placement on the right-hand-side within a linearized version of the circular EEHV1A (Kimba) genome after the actual packaging cleavage site in Raman DNA was determined [16]. However, none of the complete genomes from the GC-rich branch *Probociviruses* EEHV3A, EEHV3B or EEHV4B even have an R2-segment [19]. Note that based on the precedent from Fuery et al [7] of using the term "clades" rather than "subtypes" for the multiple E39 (ORF-Q) gene variant clusters, as well as for their corresponding ORF-Q antigens and antibodies, that descriptor was used for this particular locus only, because of its now proven usefulness for differential detection of particular EEHV1 strains in serological and immune response assays.

## Sequence sources

Publicly available nucleotide as well as protein sequences from NCBI were used for various detailed comparative analyses of our data. For the full genomic nucleotide sequences of the original five reference strains used, their case numbers and names as well as origins, year of collection and database accession numbers are listed in Table 2. Otherwise, the sources of selected relevant protein and nucleotide sequences used from incomplete sub-genomic loci are provided in the corresponding text passages or figures. Later, an expanded set of twelve fully sequenced EEHV genomes from GenBank was carried out for more quantitative assessments of their intergenomic similarity heatmap values. Similar data and accession numbers

**Table 2. Used EEHV full genomic sequences.**

| Virus | Case[1] | Individual | Origin | Year | Source |
|---|---|---|---|---|---|
| EEHV1A | NAP23 | Kimba | USA | 2004 | KC618527.1 |
| EEHV1A | EP22 | Raman | UK | 2009 | KC462165 |
| EEHV1A | EP55 | Umesh | Switzerland | 2022 | OR543011 |
| EEHV1B | EP18 | Emelia | UK | 2006 | KC462164 |
| EEHV6 | NAP35 | Miss Bets1 | USA | 2009 | MZ822422 |

[1]NAP#, North American *Proboscivirus* case number; EP#, European *Proboscivirus* case number

for those EEHV cases associated with supplementary S1 Table for both complete genomes and relevant sub-genomic PCR DNA sequence files are given in supplementary S2 Table.

## Results

**Weekly screening for EEHV-viremia.** Between January 2018 and June 2022, capillary blood of the juvenile elephants was analyzed on a weekly basis by TaqMan real-time PCR for the presence of EEHV1 and EEHV4 DNA, respectively. The results of this monitoring program are summarized in Table 3. The average Ct value for ETNF amounted to 31.5 with a standard deviation of 3.1. Although rarely and at low titers (average Ct values ranging between 35.3 and 40), transient (never on subsequent testings) and subclinical viremia by EEHV1 was detected in two of the three monitored animals. Notably, EEHV4 DNA was never detected throughout the screening period.

### EEHV serology using LIPS assay

Based on the negative cut-off value determined by horse sera, a LIPS assay using the conserved glycoprotein B of EEHV1 (gB1) as antigen, revealed that all adult individuals possessed antibodies against gB1 (Fig 2A). Interestingly, even adult African elephants reacted positively in this assay, which can be attributed to the close antigenic relationship (89% amino acid identity) of gB1 and EEHV6-gB (gB6). In contrast, the sera of two young Asian elephants from Germany who previously had succumbed to EHD did not reveal such antibodies. Interestingly, two of the Zurich elephants at risk (Omysha; Umesh) had detectable levels of anti-gB1 antibodies in their sera, whereas the third animal (Ruwani) appeared seronegative.

Next, the elephant sera were analyzed against ORF-Q clade A, B, C, and D antigens (Fig 2B-E). In general, the antibody levels against the ORF-Q antigens were considerably lower than against the conserved glycoprotein B. However, at least one individual (Indi) reacted clearly above background levels against ORF-Q clade A antigen, whereas two individuals (Omysha and Thai) were clearly seropositive against ORF-Q clade B antigen. Moreover, two of the individuals at risk (Indi, Chandra) reacted positively against ORF-Q clade C antigen. Two of the control sera (one of the German individuals who had died from EHD and one African elephant) also reacted above the background level. Although at least two individuals (Indi, Ceyla, see Fig 1) had been present during a previous outbreak of an ORF-Q clade D

**Table 3. Results of TaqMan real-time PCR targeting EEHV1, EEHV4 and ETNF.**

| Animal | Screening (weeks) | Tests (% valid) | EEHV1 | EEHV4 |
|---|---|---|---|---|
| **Omysha** | 231 | 211 (98.2%) | 6 times pos | Never detected |
| **Ruwani** | 209 | 189 (99.5%) | 3 times pos | Never detected |
| **Umesh** | 60 | 59 (98.3%) | Never detected | Never detected |

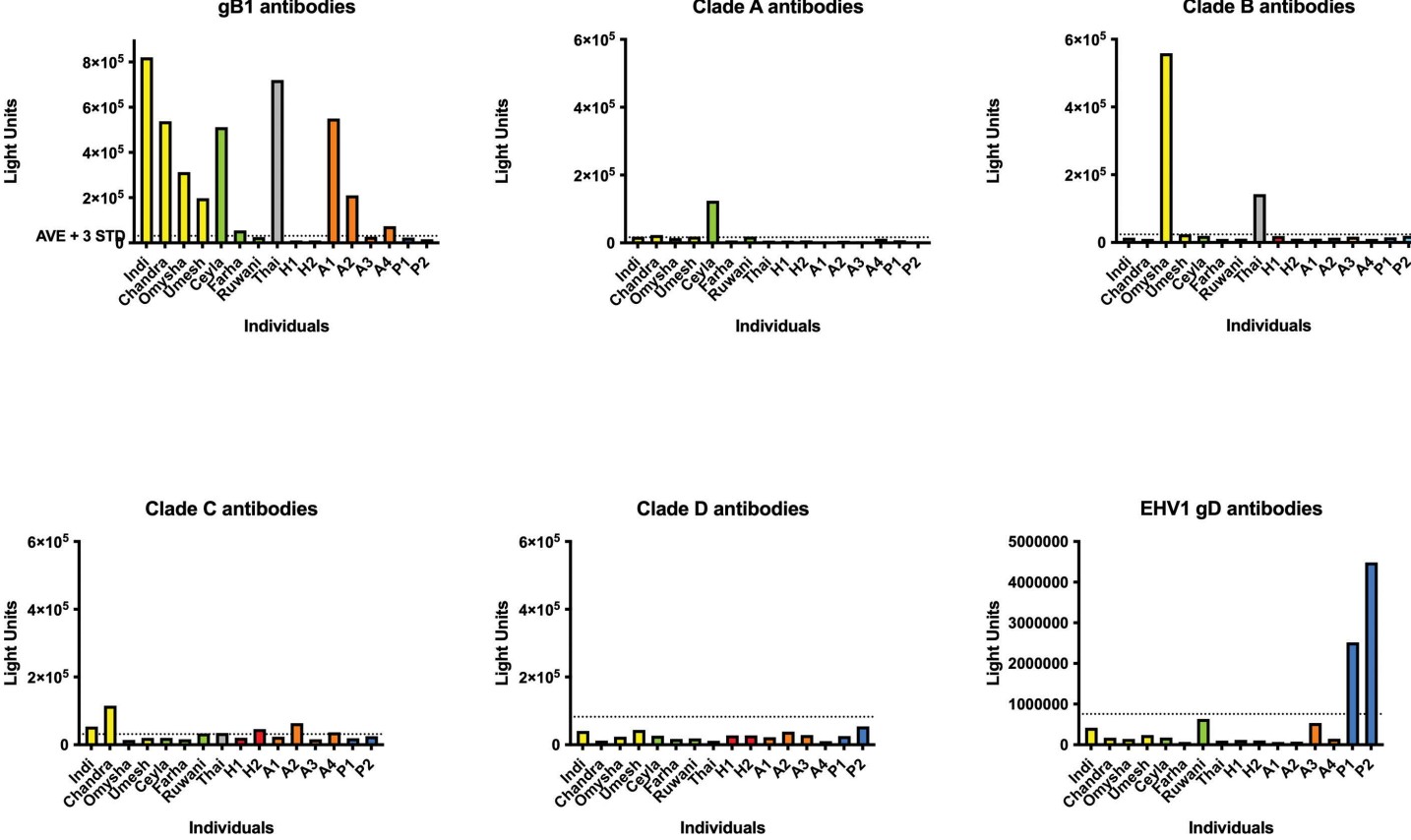

**Fig 2. Prior antibody responses against EEHV1 antigens.** LIPS assays were performed to detect antibodies against various EEHV1 antigens in the sera of elephants and horses. In each panel, the individual elephants are lined up on the x-axis, whereas the reaction of their sera is given in relative light units (RLU) on the y-axis. Yellow bars: Members of the Indi family; green bars: members of the Ceyla family; grey bar: Thai, most recent outside acquisition to the herd, introduced in 2014; red bars: sera from previous EHD cases in the Hagenbeck zoo, Hamburg, Germany; mandarin-colored bars: African elephants from the Basel zoo, Switzerland; blue bars: horse sera, which were used to determine the negative cut-off value of the test. Panel A. Antibody responses against the conserved glycoprotein gB1; **Panel B**: antibodies against ORF-Q clade A; **Panel C**: antibodies against ORF-Q clade B; **Panel D**: antibodies against ORF-Q clade C; **Panel E**: antibodies against ORF-Q clade D; **Panel F**: antibodies against equine alphaherpesvirus 1 gD.

virus (Xian, see Fig 1), none of the individuals at risk was seropositive against ORF-Q clade D antigen.

As controls, the same panel of sera was tested against the glycoprotein D (gD) antigen of equine alphaherpesvirus 1 (EHV-1). Both horse sera tested reacted positively, whereas all elephant sera had no detectable antibodies against the equine viral antigen (Fig 2F).

## Viremia upon EHD

On day -6 (see Timeframe in Materials and Mehods), all three of the monitored animals tested negative for both EEHV1 and EEHV4, but on day 1, Umesh tested EEHV1-positive with a ΔCt value of +3.8. Samples from the two following days confirmed its increasingly viremic status with ΔCt values of -0.75 and -2.75, respectively, upon which intravenous GCV treatment was initiated. Nevertheless, the ΔCt value of the same animal dropped to -4.1 and to -4.65 before coming back to +0.5 on day 6, the day of its death due to EHD (Fig 3). The other two animals remained EEHV1-negative until slight viremia was first detected on day 4 of the outbreak. Both went back to negative on days 9 and 10. On day 11, Omysha's ΔCt value was +6.15 and, after a brief recovery dropped to -1.4 (day13) before the animal died on day 18, i.e.,

10 days after initial detection of viremia. Beginning from day 8, Ruwani went forth and back with short periods of viremia. Eventually, the ΔCt also dropped to negative values (day 28), when this animal died due to EHD (day 30) (Fig 3).

## Analysis of the causative EEHV

**Sequencing of the E36/U79 locus.** A conventional PCR targeting the E36/U79 locus showed (Fig 4) that the resulting sequence obtained from an early Umesh sample was 100% identical to Xian's previous EEHV1A isolate. In contrast, 11 different nucleotides in comparison to Omysha's previous EEHV1B isolate were observed, which identified the current isolate clearly as EEHV1A.

## Initial analysis of the E54/EE51 vOX2–1 locus

Having established that Umesh had been affected by an EEHV1A strain and knowing Xian's E54/EE51 vOX2–1 sequences from 1999 (GenBank: MH287538), it was of immediate interest to determine the identity level among the two viruses. Therefore, DNA from Umesh's blood sample was amplified by conventional PCR targeting the E54 locus. Sanger sequencing of the PCR product yielded 840 nucleotides, which were compared against Xian's previously determined vOX2–1 sequence as well as against an American (Kimba, 2004) and a European EEHV1A strain (Raman, 2009) and rooted against the subtype 1B prototype (Emelia, 2006) [16,18]. On the nucleotide level, the vOX2–1 sequences of the present Umesh isolate (2022) shared only 92.34–93.84% identity with any of the compared sequences, corresponding to a minimum of 54 to a maximum of 67 nucleotide polymorphisms per sequence. The closest relationship was observed with the European 1A prototype (Raman), the least with the American 1A prototype (Kimba) (Fig 5). Overall, the present Umesh isolate differed clearly from the previous Xian isolate.

## Determination of the full viral sequences from all three cases

The full sequence of the EEHV1A virus was determined from tissue samples by next-generation sequencing. Sequences covering the complete viral genome were obtained from Umesh's samples, while the sequencing data obtained from samples from Omysha and Ruwani were less complete. The contigs obtained from Omysha and Ruwani samples were frequently interspersed by unreadable nucleotides, i.e., 27,488 unreadable nucleotides among the contigs from Omysha and 4,679 unreadable nucleotides from the contigs of Ruwani. Except for 56 ambiguous positions in the Ruwani data, the three genomes were virtually identical within the sequenced blocks. All raw sequencing data generated were uploaded to the Sequence Read Archive (SRA) as BioProject ID: PRJNA1052582.

**Overall genome structure and gene content of EEHV1A (Umesh, EP55).** The size of the assembled single complete intact genomic scaffold of Umesh (EP55) was 179,847-bp (annotated as GenBank accession number OR543011). This included two near identical copies of the 2.9-kb direct terminal redundancy (TR), one located at each end. The most distinctive features of EEHV1 genomes were all present within the Umesh genome, in particular two large-scale (20-kb and 40-kb) core gene segments, which are inverted compared to cytomegaloviruses, the putative inverted repeat stem-loop origin of lytic DNA replication (ori-Lyt, 1190-bp), the two predicted immediate-early transcriptional control proteins E40 (ORF-K, 743-aa) and E44 (ORF-L, 1209-aa), and the largest tegument protein E34 (ORF-C, 1886-aa). Other novel but consistent features found here as well as in other EEHV1 strains include the likely captured cellular genes E4 (vGCNT1) and E47 (vFUT9), plus four versions of vOX2-like proteins E23 (vOX2–4), E24 (vOX2–3), E25 (vOX2–2), and E54 (vOX2–1),

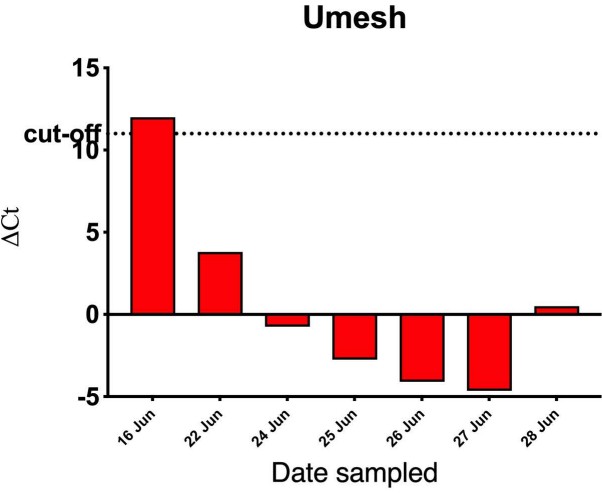

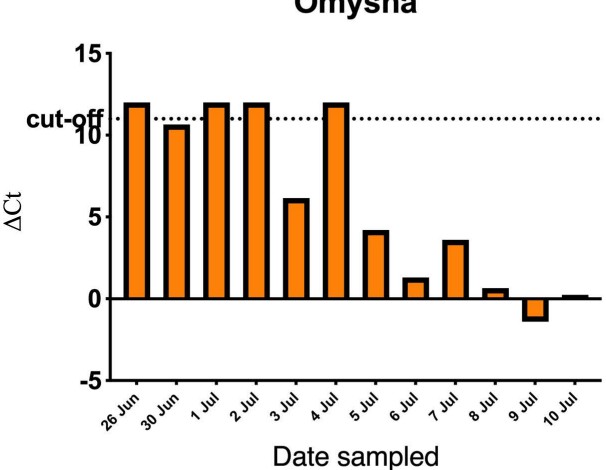

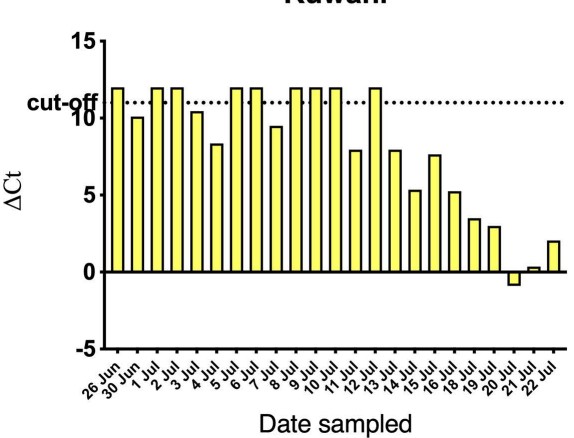

**Fig 3. Individual EEHV1 loads before and throughout development of EHD.** Delta Ct values are shown on the y-axis, whereas the date of blood sampling is given on the x-axis.

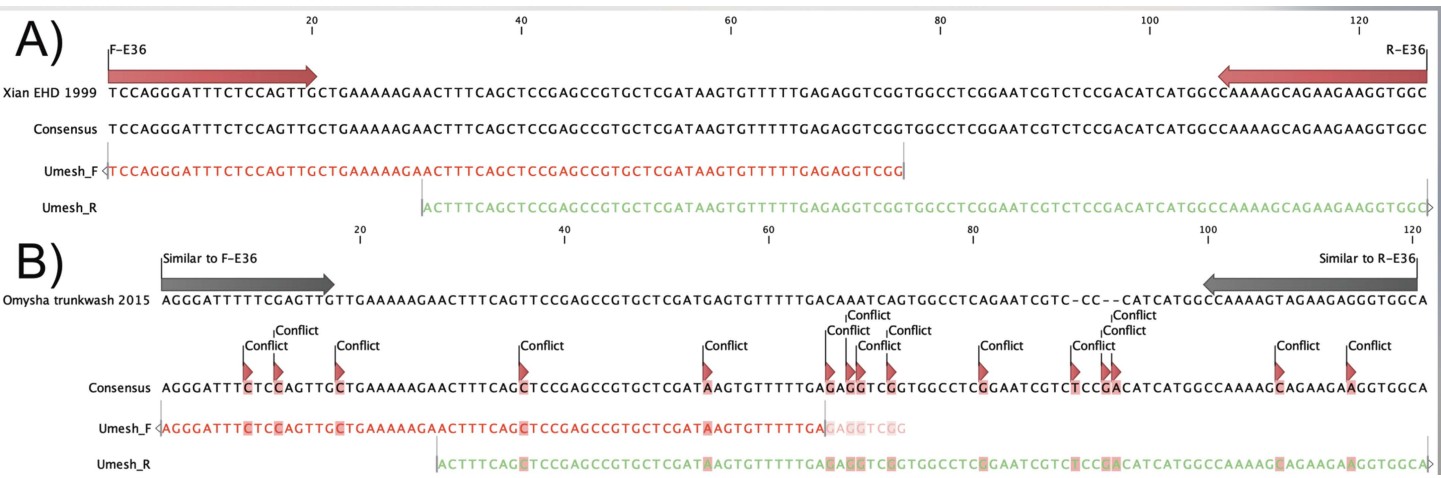

**Fig 4. Sequence alignments.** Alignment of the present E36/U79 sequence (Umesh, EP55) against previous ones. (A) Comparison against the 1A subtype (Xian, EP07). (B) Comparison against the 1B subtype (trunk wash from Omysha).

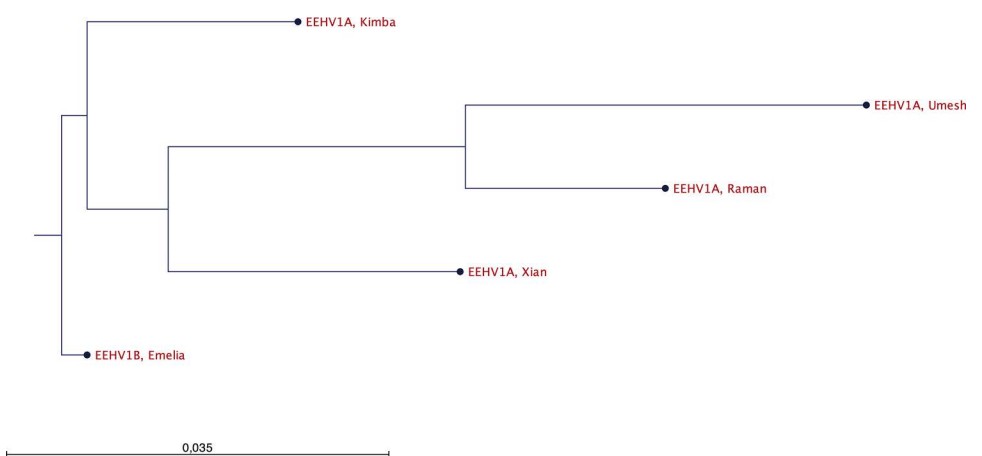

**Fig 5. vOX2–1 phylogenetic tree.** A phylogenetic tree was obtained from the alignment of the nucleotide sequences of the vOX2–1 locus (E54/EE51) from the American prototype of EEHV1A (Kimba), the European EEHV1A prototype (Raman), the European prototype of EEHV1B (Emelia) in comparison to the corresponding sequences of the present case (Umesh) and the 1999 case of EHD in the Zurich zoo (Xian).

as well as the UL9-like U73 (OBP) and both U27.5 (RRB) and U48.5 (TK) (which are all missing in cytomegalovirus genomes), and eight conserved *Proboscivirus*-specific vGPCR-like proteins, including E3 (vGPCR6), E5 (vGPCR5), E15 (vGPCR4), E20 (vGPCR4A), E21 (vGPCR4B), E26 (vGPCR3), U12 (vGPCR2), and U51 (vGPCR1) within a large family of 26 retinoic acid inducible protein 3 (RAIP3)-like 7xTM domain receptors.

## Comparison with other EEHV1A and EEHV1B genomes

Fig 6 shows the results of a search for the overall level of intergenomic DNA sequence divergence. The closest match among all ten complete EEHV1A genomes contained within the GenBank database (May 2024 version) is the Indian strain EEHV1A (IP43) [MN366291], differing from Umesh (EP55) by 2.7% at the nucleotide level (4,856-bp mismatch) as determined by direct pairwise comparisons using the VIRIDIC tool (37). It also differs from prototype

EEHV1A (Kimba) genome [KC618527] by 4.6% (9,712-bp mismatch) and from the EEHV1A (IP165) genome [MN366292] by 7.5% divergence (13,488-bp mismatch). All of the other intact EEHV1A genomes fall within the range between IP43 and IP165. For the prototype chimeric subtype EEHV1B (Emelia) genome [KC462165] from Asian elephants the divergence is 8.2% (14,747-bp mismatch) and for the orthologous prototype EEHV6 (NAP35) genome [MZ822422] from African elephants the divergence is 26.2% (47,120-bp mismatch).

Analysis of open reading frames and likely splicing motifs within the EEHV1A (Umesh, EP55) complete genome revealed that it encodes a total of 121 predicted viral proteins that are named and listed under GenBank accession numbers WNZ34470.1 to WNZ34590.1 including 17 that involve in-frame splicing. The Umesh genome proved to display similar features and gene organization as well as the typical patterns of gene content divergence, but it does not have any extra or novel genes not seen before in at least a subset of EEHV1 strains. Overall, Umesh consists of a complex mosaic pattern of mostly highly-conserved proteins intermingled together with a distinctive and unique pattern of subtypes for the 30 or so variable proteins typically found across EEHV1A genomes [1,29], but has none of the characteristic features typical of EEHV1B diaspora strains [4]. On the other hand, Umesh does contain the

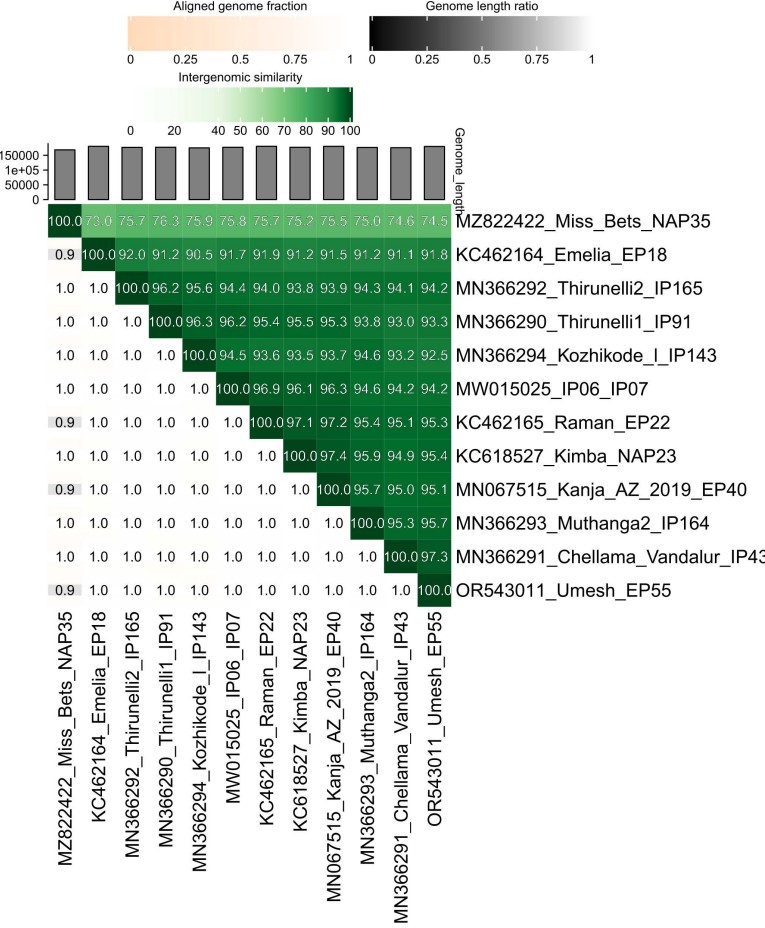

**Fig 6. Similarity heatmap.** BLAST-N alignment of the complete Umesh (EP55) genome against closely related EEHV genomes in the GenBank database using the VIRIDIC tool. The heatmap shows the similarity values for each genome pair, the fraction of genomes aligned and the genome length.

novel E36A (ORF-N) vCXCL chemokine-like protein. The latter is present in all EEHV1A strains as well as in EEHV2, EEHV5 and EEHV6, but is absent from all EEHV1B strains as well as from EEHV3 and EEHV4.

## Comparison with main segment EEHV1 variable gene subtype assignments

The EEHV1A (Umesh) assigned subtypes at each of the 20 most variable protein loci from the main segment of the genome is shown in Section I of supplementary S1 Table. The particular virus genomes chosen as illustrative examples here include the original three complete genome prototypes from GenBank of EEHV1A (Kimba), EEHV1A (Raman) and EEHV1B (Emelia), plus accumulated sub-genomic locus data from six of the best characterized but incomplete EEHV1A genomes determined previously by conventional PCR amplification-based DNA sequencing of older lethal EHD cases as carried out by the Johns Hopkins University Viral Oncology group. Specifically, this data was derived from one sample from India (IP11), one from Europe (EP07, Xian) and four from the United States (NAP18, NAP26, NAP17 and NAP21).

Other key details about these ten EEHV1 strains are listed in S2 Table, including where appropriate GenBank accession numbers for complete genomes as well as for six selected representative sub-genomic loci. With the exception of the prototype EEHV1A (Kimba) strain, which (although it does have three Cs within CD-II) was otherwise designated to have just A or A1-subtype variable genes across the entire genome, all other non-1B diaspora strains fall into complex mosaic patterns like this with largely unlinked variable gene subtypes.

The combined ORF-O/ORF-P/ORF-Q complex (part of CD-III), for which we recognize a total of thirteen different inserted alternative three-gene patterns, with the same C-E-C subtypes as in Umesh are found also in eleven out of a total of 52 other EEHV1A or EEHV1B strains. Those include the viruses from European cases Raman and Plai Kiri as well as from IP43, NAP16, NAP24, NAP43, NAP73, NAP75, NAP88, NAP91 and NAP93. Although only two of those examples are available as complete genomes (Raman and IP43), none of them occur within known EEHV1B strains. In addition, only one of the seven India cases evaluated here (IP43) has that same C-E-C group pattern.

## Comparison with other EEHV1 R2-segment gene arrangements

Similarly, many of the eleven potential protein coding positions located within the short 10.5-kb R2-segment of all EEHV1 genomes display high levels of variability and subtyping that have proven to be unrelated to the linked 1A/1B chimeric domain pattern explained above. These latter non-1A/1B diaspora related hypervariable gene loci all map within the novel *Proboscivirus*-specific segments located near both ends of the linear EEHV1 genomes. Although the R2-segment mapping between E47 (vFUT9) to E55 (vIgFam3) encompasses just 5% of the length of the EEHV1 genome, it has alternative three-gene "cassettes" that define eight distinct clustered R2 sub-groups designated A, B, C, D, E, F, G and H. Each strain carries just one of these triplex cassettes and there are three different potential locations where they can be inserted. Therefore, these clustered groupings represent major determinants of EEHV1 genomic organization.

Unlike the 20 known available "optional" R2-family genes present within the cassettes, which are each found in just a small subset of the overall EEHV1A strain population, there are also eight remaining "constant" R2-segment genes (see S1 Table) that are almost always all present but are also mostly highly diverged and form many additional subtype clusters of their own. The latter include two captured host genes E47 (vFUT9) and E54 (vOX2–1) plus six members of multigene paralogous R2-segment gene families.

Although each individual EEHV1 genome usually contains a total of just eleven genes (often including an average of two or three fragmented or inactivated ORFs) within the entire R2-segment, the overall population of 38 distinct independent EEHV1A plus EEHV1B virus strains that have been evaluated fully across the R2-segment encodes a total of 28 different available alternative genes. The presence (+) or absence (-) as well as the relative location and subtype designation of 20 of these alternate genes found within the R2-segments of either Umesh itself or one or more of the set of nine reference EEHV1 genomes are summarized in Section II of the supplementary S1 Table. Note that only four of the total of eight known organizational patterns of alternative R2-segment insertion cassettes are represented here (i.e., A, B, D and F groups, but not any examples of the C, E, G or H groups).

These other R2-segment genes include seven more alternative versions of vGPCR-receptors, namely E48 (vGPCR7), E50.5 (vGPCR8/vGPCR9/vGPCR13), E56 (vGPCR12), E59 (vGPCR10 and E62 (vGPCR11) from within the large 7 x TM RAIP3-like family that each occurs in just a subset of strains. Every EEHV1 R2-segment includes either just one or two of the latter with Umesh itself having versions of both E50.5 (vGPCR9) and E59 (vGPCR10) but not any of the others. There are also a total of 18 paralogous diverged alternative versions of mostly membrane-bound immunoglobulin-domain (Ig) family proteins (designated vIgFam1 to vIgFam15) distributed amongst known EEHV1 strains. Umesh encodes its own versions of five of these, including three that are constant, e.g., E50 (vIgFam1), E53(vIgFam2.5) and E55 (vIgFam3) plus two of the alternates E60 (vIgFam6) and E61 (vIgFam7). Some of these are detectably related to host cell protein CD48. Finally, there is also a small third family of four other rather non-descript alternative R2-segment membrane proteins (E49, E51, E64 and E68).

The Umesh pattern of R2-genes most closely matches that of the Ganesh (NAP26) and IP43 by having a typical group-D E59/60/61 three-gene triplex cassette among the total of eleven R2-segment encoded proteins overall (see S1 Table). Unlike the other R2 genes described here, three of the vIgFam proteins (E60, E61 and E53) do not show any significant subtype variability, whereas in contrast the 43 examples of E54 (vOX2–1) proteins evaluated, despite collectively exhibiting up to 15% overall variability at the aa level, do not cluster into any readily definable subtype groupings. In contrast there are just two tight major subtype clusters of E47 (vFUT9) in which the prototype EEHV1A (Kimba) version (subtype-A) and the prototype EEHV1B (Emelia) version (subtype-B) differ by as much as 32% at the protein level. Nevertheless, this locus shows no obvious linkage to the classic 1A/1B-subtype diaspora. In fact, from among the 35 total E47 (vFUT9) enzyme-encoding versions evaluated here there are 24 examples of the B-subtype (including Umesh) and only eleven of the A-subtype, with both being randomly assorted among EEHV1A and EEHV1B diaspora genomes.

Overall, the eight different alternative three-gene inserted cassette groups provide the most definitive way to classify EEHV1 genome patterns within the R2-segment. Here the Umesh strain clusters within the largest set (= group-D) along with ten other group-D examples (namely Daisy, IP43, Xian, EP20 = EP21, NAP30, NAP31 and NAP41 = NAP47), including three from Europe (Xian, EP20 = EP21), but just one from India (IP43). Moreover, there are no known group-D inserted cassette patterns (i.e., E59/E60/E61) that fall amongst the six EEHV1B genomes for which R2-segment data is available. The next two most abundant cassette patterns are seven group-As and six group-Bs, with the remaining 15 examples being three -Cs, two -Es, four -Fs, two -Gs and four -Hs. Among the latter, there are several other anomalous situations where a total of ten virus strains display either just a two-gene cassette (groups-C, -E and -G) or four others have both a group -B like three-gene cassette plus a two-gene group -H cassette (not shown).

Section III of S1 Table includes examples of subtype designations applied previously to five core conserved genes from these and many other EEHV1 genomes based on selected small PCR-sequenced loci including E38 (POL) that just display low nucleotide level variability without significant associated protein level variation [1,3,4,34].

These detailed DNA sequencing results confirmed that all three EHD cases here had been infected by exactly the same novel strain of EEHV subtype 1A.

## Further analysis of the Umesh E39/EE3 locus and comparison to other ORF-Q subtypes

Antibodies against the same particular ORF-Q clade as the infecting strain have previously been associated with a mild course of EEHV1 infection, whereas their absence has been linked to severe EHD (7). An updated ORF-Q phylogenetic tree was generated, showing the alignment results for the intact translated protein sequences from 33 independent strains (Fig 7). The year 2022 Umesh isolate proved to be nearly identical to other ORF-Q clade C isolates such as Plai Kiri and Raman (plus NAP43, 73, 75 and IP43), but highly diverged from clade A (NAP11, 16, 18, 23, 26, 32, 39, 41, 47, 80; IP11, 164), clade D (Xian, EP21; NAP20, 29, 30, 31, 72; IP07) and clade B (NAP45, 49) strains as well as from clade E1 (NAP21), E2 (IP91, 165) and F (NAP17), and further still from EEHV6 (NAP35). Quantitatively, the average percentage protein identity values when compared against the 364 aa version encoded by Umesh as measured by the NCBI BLAST-P program proved to be 98–99% for the other members of clade C, 77–82% for clade A, 45–47% for clade D, 30% for clade F, 28% for clade E2, 26% for clade E1, 26% for clade B and 33% for EEHV6 (NAP35), whose sequence was used as out-group for rooting the tree. These epidemiologically relevant data indicate that a previously

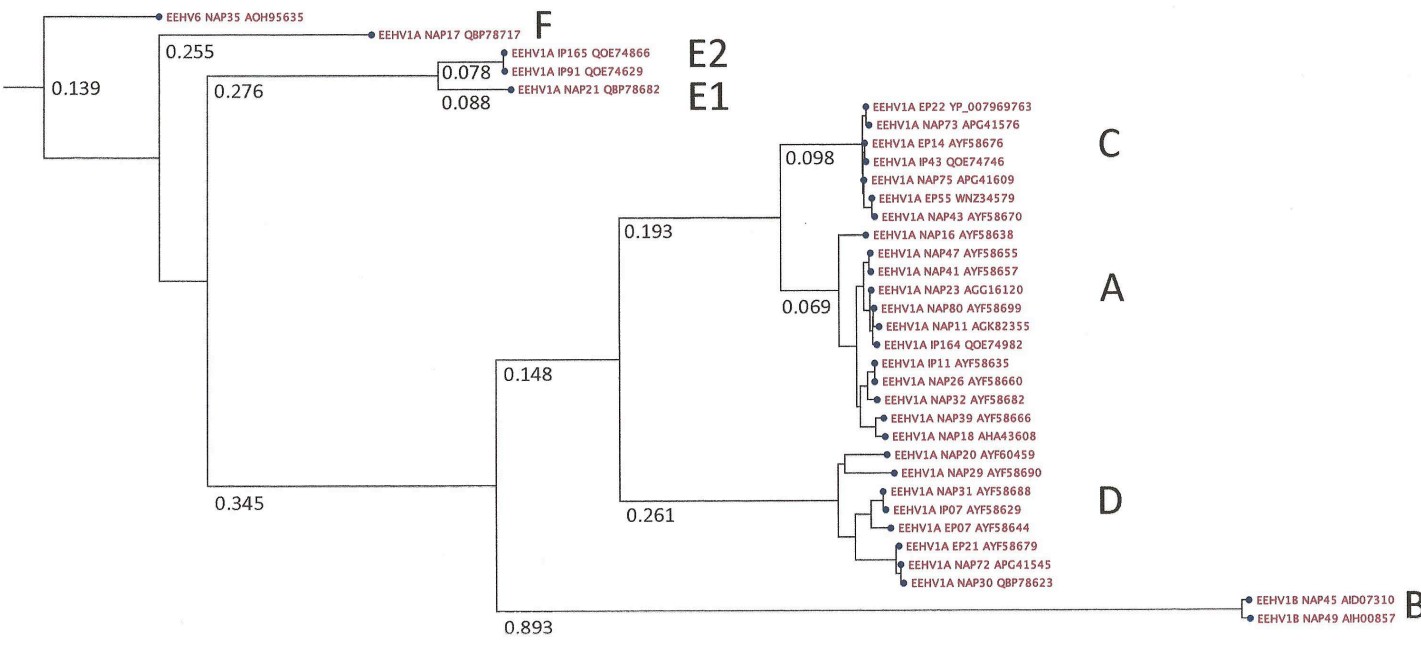

**Fig 7. ORF-Q phylogenetic tree.** A maximum likelihood phylogenetic tree was obtained from the alignment of ORF-Q amino acid sequences using the Neighbor Joining method and 100 replicates to perform bootstrap analysis. The branch length is indicated for major branches, dividing the tree into the known clades A through F. Each strain is identified by its case number followed by its GenBank accession number.

unrecognized, newly detected strain of EEHV1A had been the cause of the present EHD outbreak.

### Analysis of the U48.5/EE7 ETk locus

As the ETk may contribute to the success of antiviral treatment with nucleoside analogues, it was also of interest to compare the predicted protein sequences of Umesh ETk against previously known EEHV1 ETks. Indeed, the aa sequence was highly conserved among reference EEHV1A isolates, with only one aa difference (H278N) against the sequence from the European EEHV1A prototype (Raman) and two aa (V152I and H278N) against the American prototype (Kimba).

### Analysis of the U69 ECPK locus

The aa sequences of Umesh ECPK are highly conserved when compared to other EEHV1A strains and were identical to the sequences of the European EEHV1A isolates Raman or Plai Kiri. One single aa exchange (T409A) was detected upon comparison against the EEHV1B isolate Emelia. The ECPK sequence of Kimba contained two aa differences, namely T408P and T409P.

### Analysis of the U38 DNA polymerase locus

The viral DNA polymerase enzyme (POL), encoded by U38, is essential for viral DNA replication. Phosphorylated nucleoside analogues may either block the enzyme, thereby terminating its activity, or integrate chain-terminating molecules into the growing DNA, which blocks further elongation. The sequence conservation among all three isolates at this locus was 100%, whereas two to eight aa changes were observed upon comparison of the Umesh sequence against those from ten previously published cases. While seven of those differences had been observed previously, Leu 792 occurred in our sequence compared to Isoleucine at this position in all ten previously published isolates (L792I). This seeming mutation mapped to the center of a conserved domain, which had been associated with predicted enzymatic activities, including elongation, DNA-binding and dNTP binding.

## Discussion

### Detailed knowledge of circulating viruses may influence treatment options

With a history of two prior cases of EHD at the zoo and evidence of recurrent episodes of EEHV shedding and viremia between 2014 and June 2022, the risk for another outbreak of EHD was considered high. In particular, the prevalence of EEHV1A, EEHV1B, and EEHV4 among the adult animals, and the presence of two seronegative animals in the susceptible age were documented [25]. During four years of weekly screening, seven instances of low grade, transient viremia were detected, each caused by EEHV1, while viremia due to EEHV4 was never detected (Table 3).

EEHV serology has only recently established its proper role in addressing EEHV prevalence among elephants [2,7,14]. Particularly LIPS assays have proved to allow us to discriminate between antibodies that cross-react among various EEHV types or strains and other antibodies that represent a specific immune response against a particular virus strain (7). We selected LIPS assays using the conserved, cross-reactive glycoprotein B- (gB) and the subtype-specific ORF-Q-antigens, respectively, to assess the serological status of the animals against EEHV. The use of ProteinA/ProteinG-beads in the LIPS assays brings a great advantage for the veterinary medical purpose with it because different animal species can be

addressed using the same assay. We took this advantage in our case by using horse sera to determine the negative cut-off values for all of our LIPS assays.

All adult elephants had detectable levels of antibodies against gB of EEHV1A (gB1A), which indirectly confirms active circulation of EEHV1 among the elephants. Notably, sera from four African elephants, which were provided for control purposes by the Basel zoo, also yielded varying degrees of antibodies against gB1A. The published protein sequence of the EEHV6 gB (gB6) shares 88–89% aa identity with gB1 of the subtype 1A strains from Kimba and Raman. The protein sequence of gB subtype 1B (Emelia) shares only about 85% aa identity with either EEHV6 (NAP35) or EEHV1A (Kimba, Raman). Consequently, we assume that the gB-antibodies detected in the African elephants were probably a result of cross-reacting subclinical infection with EEHV6.

At eight years of age and with a history of shedding EEHV1B [25] as well as several occasions of transient viremia (Table 3), the seropositive status of Omysha could be attributed to having mounted her own active immune response against EEHV1. At two years of age and without previously detected episodes of viremia, the seropositive status of Umesh may be explained by low maternal antibodies, but as half-brother and playing companion to Omysha, it is possible that this individual had already mounted its own immune response against EEHV1 under the previous cover of maternal antibodies. The seronegative status of Ruwani at the age of five years was a matter of concern. No traces of maternal antibodies could be detected, and an active immunity seemed not to have developed, although three episodes of viremia had previously been recorded (Table 3).

LIPS assays using ORF-Q antigens detected fewer seropositive animals than gB-serology. As the subtype EEHV1B had been observed to be shed from the trunk of Omysha (25), it was not surprising that this animal showed the highest reaction among the members of the herd against the ORF-Q-B antigen. In contrast, the other young animals at risk did not have such antibodies, nor were they seropositive against ORF-Q antigens A, C, or D. Of the older members of the herd, at least one female elephant (Ceyla) had significant antibody levels against ORF-Q-A, whereas two female elephants (Indi; Chandra) had preexisting antibodies against ORF-Q-C. Interestingly, none of the animals seemed to have significant levels of antibodies against ORF-Q-D antigen, although the isolate from a previous case of EHD in the zoo (Xian, 1999) had been classified to belong to ORF-Q clade D. These results suggested that antibodies against ORF-Q antigens may be short-lived and confirm that young elephants without matching ORF-Q antibodies are at risk during an EEHV outbreak [7].

Thanks to the preventive viremia-screening program, the 2022 outbreak was detected early. However, due to the "protected contact" principles, any treatment option depended on the individual elephant's decision to interact [28]. Although Umesh agreed to intravenous GCV treatment (details to be published elsewhere), the virus burden in the bloodstream increased dramatically (Fig 3) and six days after viremia was first detected, Umesh died of EHD. The second animal (Omysha) rebounded twice from viremic to aviremic and back but then still succumbed to EHD. Notably, Omysha did not agree to GCV treatment but consented occasionally to take up some oral dose of FCV. Neither GCV nor FCV had a noticeable effect on the ongoing viremia (Fig 3). The third animal (Ruwani) had its first positive reading of viremia at the same day as the second animal (Omysha) but had a survival time of three weeks after that, even without consenting to any treatment option. Only after its fifth detection of viremia, the virus burden in the blood stream increased with almost every consecutive sample and the animal eventually succumbed to the disease.

Once the outbreak was ongoing, it was of immediate interest to subtype the causative virus more closely. Indeed, initial PCR-sequencing of the E36 gene locus suggested the involvement of a subtype 1A strain, rather than a subtype 1B virus as the current disease-causing

agent. This conclusion raised new concern about the eight-year-old Omysha, which had been found seropositive for antibodies against the ORF-Q-B antigen but not against other ORF-Q antigens. Moreover, the possibility to use her antibody-rich serum for a potential treatment of the other animals at risk was now ruled out. As maternal antibodies seem to be able to confer protection, a rapid analysis of the prevailing virus subtype may help to identify quickly a suitable herd mate as serum donor [7].

Subsequently, it was inviting to speculate that one of the previously detected 1A strains, dating back to 1999 and 2003, had re-emerged after long years of latency, which had been the case in Berlin, with Shaina Pali and Ko-Raya (EP23 and EP25) being identical to Plai Kiri from 11 years earlier (GenBank KT705234.1, KT705238.1 and KT705239.1)(1). Yet, a comparison of E54 sequences, including the nucleotide sequences of the Xian (1999) isolate and Umesh revealed significant differences. Thus, the present EEHV1A virus was newly detected throughout this outbreak, ruling out the re-emergence of a previously identified latent strain. These analyses during the ongoing outbreak had direct effects on our risk assessment and may have also expanded our treatment options. Identification of the ORF-Q subtype of the prevailing virus had not been established at the time of the outbreak and ORF-Q serology using subtype C and D antigens were only established in retrospect. Otherwise, Indi and Chandra, both seropositive for ORF-Q-C antibodies (Fig 2), may have been used as serum donors for antibody transfer.

## Evidence that the same new EEHV1A strain infected all three calves

Only once the outbreak of EHD was over, could the entire EEHV1 genomes from all three cases be determined and compared with the result of the initial PCR analysis. Moreover, the sequences of other genomic loci of interest then became available for interpreting the course and origin of the disease, as well as the failure of treatment.

Concerning the epidemiological chain, it is important to re-emphasize that all three isolates had the exact same genomic sequence, belonging to the EEHV1A subtype. The E54 vOX2–1 locus was 100% conserved among the three current isolates, which confirmed their common origin, but was significantly different from the Xian (1999) isolate. Furthermore, analysis of the E39 locus of Umesh identified a virus belonging to the ORF-Q-C clade. This finding was very surprising, since no such virus had previously been identified among these animals. Instead, the previous lethal EEHV1A strain responsible for the death of Xian, belonged to the ORF-Q-D clade, while serological evidence suggests that the EEHV1B isolate detected in earlier trunk-wash samples from Omysha belonged to the ORF-Q-B clade. These observations are consistent with an earlier report, suggesting that the presence of clade matching antibodies against the ORF-Q antigen of the infecting virus was associated with protection against death from EHD [7].

The ORF-P and ORF-Q genes of EEHV1, which arose by duplication and have some residual similarity, are replaced in EEHV3/4 by a single novel gene at the same location and with the same two exons (E39A, ORF-R), but with minimal or no homology to ORF-P or to ORF-Q itself [5]. Similarly, EEHV2 and EEHV5 also lack ORF-Q but do have ORF-P. Accordingly, our serology was unable to evaluate the serological status of the animals towards EEHV3/4, although EEHV4 had been detected in the trunk wash sample from Thai the breeding bull, and LIPS assays for the specific detection of antibodies against EEHV2 and EEHV3/4 have recently been established by others on the basis of the E34 (ORF-C) encoded protein [15]. Unfortunately, the ORF-Q sequence from Aishu, who died from EHD in 2003, was not available. However, the few known parts from the Aishu virus genome, for example the U48.5 ETk locus, differ considerably from the present sequences. Overall, these sequences confirmed the presence of a new previously undescribed virus strain among these elephants.

## Potential antiviral drug targets

Three different EEHV1 encoded enzymes may be involved in the susceptibility or resistance of nucleoside-based antiviral treatment against EEHV: the viral thymidine kinase (U48.5, ETk), the conserved protein kinase (U69, ECPK), and the catalytic unit of the viral DNA polymerase (U38, DNA pol) [17,39–41].

Identities and differences, respectively, in the coding sequences within these three genes might not only convey information about potential drug-resistance but also about the origin and sequence of transmission. Since all sequences from our three isolates were identical, the source of infection is either the same for all three animals, or the virus must have been transmitted from one case to the other without genomic alteration, with all three animals effectively being infected from the same original shedder.

As its name suggests, the ECPK protein was even more conserved than ETk. Moreover, this enzyme did show some *in vitro* activity, converting GCV and PCV in a dose-dependent manner, thus, making it a prime candidate not only for rendering the virus sensitive to those antiviral drugs, but also for a possible emergence of a drug-resistant mutant virus [1,40,41]. Although a few variations against previous sequences were detected, all the differences had been identified before and mapped outside of the predicted catalytic domain of the enzyme (aa 166–306). In the present case, three different aspects have to be considered, which may explain the obvious inability of the antiviral drugs to confer any measurable effect on viral burdens, as well as on the clinical outcome of the disease: [1] ECPK and ETk are less active than HSV Tk in converting the prodrugs to their active form, which may negatively affect the desired drug activities, and [2] it is possible that the U38 DNA polymerase may be naturally incompetent to incorporate certain phosphorylated nucleoside analogues into the growing viral DNA. Because of the lack of suitable cell cultures, this issue has not yet been addressed. Still, an interesting variation was observed within the U38 DNA polymerase sequence, where the same single mutation (L729I) occurred in all three of our current new isolates as opposed to the high conservation of a different amino acid here among ten randomly selected previous isolates. This possible mutation mapped to the center of a conserved domain, which had been associated with predicted enzymatic activities, including elongation, DNA-binding, and dNTP binding. [3] Because the treatment efforts depended on the affected animal's cooperation, it is also possible that doses of antiviral drugs did not meet the needed concentrations in the bloodstream.

## Presumed long-term quiescence in a herd-mate

Having established that all three animals had been infected most likely from the same source with the same but previously unrecognized virus, its most probable origin can be attributed to an elephant who was born outside of the Zurich zoo. Indeed, ORF-Q serology identified two individuals, Indi and her daughter Chandra, as the only ones that had been ORF-Q-C-seropositive prior to the outbreak. Chandra had been born and raised in the zoo, whereas her mother (Indi) had been born 1986 in Myanmar, and was imported to Switzerland in 1988, before being transferred to the Zurich zoo in 1999. One could speculate that Indi has acquired this "novel" virus in her youth but did not spread it until prior to the present outbreak.

## Global protein subtype analysis of the intact genome of EEHV1A (Umesh, EP55)

The large DNA genomes of EEHV1 and the six other identified *Proboscivirus* species (EEHV2 to EEHV7) range between 168 and 206-kb in size encoding approximately 120 genes each and have amongst the most variable DNA genomes of all mammalian herpesviruses. This extreme

hypervariability is largely confined to eight chimeric domains (CD-I to CD-VIII), as well as the variable gene content within the 4–10-kb R2-segment. The hypervariable domains of the most variable EEHV1 proteins often cluster tightly into three to eight distinct cladal patterns that have been designated as subtypes A through H. In the special situation of the chimeric EEHV1B genomes, the individual gene subtype patterns of the non-adjacent CD-1, CD-II and CD-III variable loci across large sections of the genome are linked, but there is very little linkage observed between the subtype patterns across different EEHV1A strains. Here in the case of the lethal EEHV1A (Umesh, EP55) strain that infected three young Asian elephant calf mates within just a few days of one another, we have determined the complete 175-kb genomic DNA sequence with full annotation of the gene coding content and compared it with nine other selected representative well-studied EEHV1 genomes in supplementary S1 and S2 Tables). In particular, we have focused especially on the designated subtypes of all of the 33 most significant hypervariable viral proteins including those contributing to the unique R2-segment gene content. The combined individual protein subtypes obtained for EEHV1A (Umesh) conform to the general expectations of a novel overall mosaic pattern with considerable subtype level divergence across just a scattered subset of hypervariable proteins compared to all other evaluated strains. For Umesh this includes falling into the D-subgroup R2-segment cluster together with two other reference strains (Xian and Ganesh) as defined by the presence of the adjacent E59 (vGPCR10) plus E60 (vIgFam6) and E61 (vIgFam7) genes as their characteristic alternative inserted R2-triplex cassettes. Among the particular ten reference EEHV genomes evaluated in S1 Table, Umesh is also the only one to encode an E-subtype U51 (vGPCR1) protein as well as the only one to encode a D-subtype U48 (gH) protein. It also shares the C-E-C subtype pattern for the ORF-O, ORF-P, ORF-Q main segment glycoprotein complex only with EEHV1A (Raman).

The fact that EHD evidently involves primary infection in inappropriately or insufficiently immune calves [7,8,15] appears fully consistent with the universal finding of the identical species and strain of EEHV infecting two (or more) elephant calves that contracted DNA-positive confirmed EHD nearly simultaneously at the same housing facility. These occurrences have now been documented at least a dozen times previously, including in Germany and multiple times in the United Kingdom and USA, plus three times also in Asian range countries India [3,29], Indonesia [42] and Malaysia [43]. Only in Switzerland and Malaysia, however, have the episodes involved as many as three separate lethal EHD cases within young calves with the identical virus strain within a very short time period.

## Conclusions

Despite considerable preventive efforts and early detection of EEHV1A viremia, the Zurich zoo lost three young elephants to EHD. The reasons for the failure of the preventive measures may be summarized in three points: [1] Presence of a previously undetected EEHV1A strain among the resident elephants. Possibly, it could be deduced that the new virus re-emerged after almost 40 years of latency from one of the oldest elephants in the zoo. [2] Insufficient or absent strain-specific immunity in the affected animals. [3] Lack of proven efficacy antiviral drugs and protective vaccines, due to the unavailability of appropriate cell cultures and animal models. Having determined the complete genomic DNA sequence of EEHV1A (Umesh, EP55) during the course of this outbreak, we also took the opportunity to make detailed comparisons of the overall gene content and hypervariable protein subtype clustering patterns with multiple other EEHV1 strains leading to a new level of appreciation of the often very high levels of polymorphic divergence within a subset of the proteins encoded among and across these genomes, especially within the R2-segment. The complexity of those differences with major and minor variations has to be accounted for in the planning of future strategies

for prevention and treatment of EHD. Based on the obvious conservation of the three enzymes responsible for sensitivity and resistance to nucleoside analogues, it seems to be more straightforward and feasible to develop efficient antiviral drugs than to understand the role of all those proteins encoded in the chimeric domains and the R2 segment, which may have immunologically decisive effects on conferring protection against EHD upon immunization.

## Supporting information

**S1 File. Comparison of EEHV genomes.**
(DOCX)

**S1 Table. EEHV1A (Umesh) Variable Gene Subtype Comparisons.**
(DOCX)

**S2Table. Details about the Ten EEHV1 Genomes Used in Table S1.**
(DOCX)

## Acknowledgments

We are grateful to the zoos in Zurich, Basel, and Hamburg for their contributions. Equally, Dr. Monika Hilbe and Prof. Dr. Franco Guscetti, Institute of Veterinary Pathology, University of Zurich are greatly acknowledged for providing necropsy tissue. We also thank Prof. Dr. Cornel Fraefel for his constant support of the viremia screening and antibody testing. We thank Dr. Andreina Schramm for sharing her EHV1 positive horse reference sera. We also acknowledge the direct collaboration and assistance of Erin Latimer at the Elephant Herpesvirus Laboratory of the Smithsonian National Zoo and J.-C. Zong of the Viral Oncology Program at Johns Hopkins School of Medicine for much of the unpublished sub-genomic PCR sequencing analysis work carried out on earlier EHD cases collected worldwide that was drawn upon from GenBank accession files for the strain differences analyses provided here in Supplementary Material S1 and S2 Tables.

## Author contributions

**Conceptualization:** Mathias Ackermann, Gary S. Hayward.

**Data curation:** Sarah Heaggans, Gary S. Hayward.

**Formal analysis:** Mathias Ackermann, Jakub Kubacki, Sarah Heaggans, Gary S. Hayward.

**Investigation:** Mathias Ackermann, Jakub Kubacki, Gary S. Hayward, Julia Lechmann.

**Methodology:** Mathias Ackermann, Jakub Kubacki, Gary S. Hayward, Julia Lechmann.

**Validation:** Gary S. Hayward.

**Visualization:** Mathias Ackermann, Jakub Kubacki, Gary S. Hayward.

**Writing – original draft:** Mathias Ackermann, Jakub Kubacki, Gary S. Hayward, Julia Lechmann.

**Writing – review & editing:** Mathias Ackermann, Jakub Kubacki, Gary S. Hayward, Julia Lechmann.

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
