## [Decision Letter · Decision Letter 0]

24 May 2024

PONE-D-24-09462Epidemiological, Serological, and Virological Analysis of an Outbreak of Elephant Hemorrhagic Disease in SwitzerlandPLOS ONE

Dear Dr. Ackermann,

Thank you for submitting your manuscript to PLOS ONE. After careful consideration, we feel that it has merit but does not fully meet PLOS ONE’s publication criteria as it currently stands. Therefore, we invite you to submit a revised version of the manuscript that addresses the points raised during the review process.

We look forward to receiving your revised manuscript.

Kind regards,

Vishwanatha R. A. P. Reddy

Academic Editor

PLOS ONE

Journal Requirements:

3. Please note that your Data Availability Statement is currently missing a direct link to access each database. If your manuscript is accepted for publication, you will be asked to provide these details on a very short timeline. We therefore suggest that you provide this information now, though we will not hold up the peer review process if you are unable.

Reviewers' comments:

Reviewer's Responses to Questions

**Comments to the Author**

1. Is the manuscript technically sound, and do the data support the conclusions?

Reviewer #1: Yes

Reviewer #2: Yes

2. Has the statistical analysis been performed appropriately and rigorously? 

Reviewer #1: Yes

Reviewer #2: Yes

3. Have the authors made all data underlying the findings in their manuscript fully available?

Reviewer #1: Yes

Reviewer #2: Yes

4. Is the manuscript presented in an intelligible fashion and written in standard English?

Reviewer #1: Yes

Reviewer #2: No

5. Review Comments to the Author

Reviewer #1: Thank you for sharing our data - this is a good addition to the current knowledge availabe.

Please find the comments on your paper in the attached.

Please consider a major revision, especially of the results section (abridge), and move excessive explanations to suppl materials.

Reviewer #2: PONE-D-24-09462

EEHV outbreak[s] in Switzerland

Elephants are hosts to several closely related herpesviruses known as Elephant endotheliotropic herpesviruses (EEHVs). These viruses can cause Elephant hemorrhagic disease (EHD), which is frequently fatal, and poses a risk to the precious and carefully managed herds of elephants that live in zoos around the world.

As a group, EEHVs have been classified as members of the betaherpesvirus subfamily, but they have genetic properties that make them distinct from other herpesviruses. Based on their DNA sequences, isolates of EEHV have been informally classified into several types, e.g., EEHV1A. Within types, strain-specific sequences have been detected. At present, the available information has been insufficient to motivate formal action related to classification of EEHV strains into distinct virus species.

This manuscript describes a very sad story about an outbreak of EHD that killed three young elephants in a Swiss zoo.

The paper is important because it is a carefully detailed description of the outbreak and the actions taken to understand and respond to what happened. It is important that this thoughtfully compiled information be peer-reviewed and made available in a public archive, such as PLOS One, as a resource for whoever needs to deal with such a problem in the years to come.

This work is likely to have long-term positive impact, but probably not of the sort measured by citation frequency.

Suggested modifications:

L30: “This virus probably re-emerged” (it is not a “new” virus”)

L73: “knowledge of …is also important”

L80. Define “ECPK”

L85: delete “at least”

L103: “to admit”

L183-186. A real-time PCR assay was used to obtain relative quantitation of virus and host genome copy numbers. The virus target is present at one copy per virus genome, and the host target is present at two copies per diploid genome. The text should be modified to make this clear, as well as its influence on data interpretation.

L315. Is “at least” needed here?

L403. I do not know the meaning of “27’488 and 4679 NNNs”.

L405. A denominator is needed for the 56 nt mismatch.

L417. Not sure what is meant by “non-core” likely tegument protein.

L420. Roseoloviruses also encode OBPs.

The L417 and L420 items end up being mentioned in lines 431-434, but those areas might be able to be more tightly integrated.

L448. Is the hypervariability within isolates or strains, or between isolates or strains?

The complex string spanning lines 509-513 contains what seem to be inconsistencies in usage of commas, semicolons, periods, and parentheses.

L528. p. 22. Not sure what is meant by “triplex patterns”, which are mentioned here and on p. 22 line 528.

L642. “routing” to “rooting”

L670. “sequence of Kimba contained”

L672. “address their biological function in the future”

L675. delete “considered”

L676-677. “enzyme, thereby terminating its activity, or”

L682. I don’t understand the sentence that begins with “Interestingly”

L695. delete “single one”

Having defined its abbreviation in line 702, use “gB” thereafter.

L713. “of gB subtype 1B”

L728. “revealed much” to “detected”

L762. Is “apparently” needed?

L778. “homology” to “similarity”

L786. “new” to “previously undescribed”

L792. “convey information about”

L793. “If all” to “Since”

L802. “Despite the long”

Lines 805, 807, and 809 say the same thing three different ways, which is confusing. “U38 DNA polymerase”, “U38 protein”, and U38 DNA pol. Pick a winner.

Other comments:

1. It seems like Umesh’s sequence should be included in Table 2.

2. In Fig. 1, the orange used for Thai, is too similar to Indi’s yellow for the parentage of Umesh to be unambiguous. A similar situation exists for Ruwani. It would be helpful to indicate which animals died from EHD. Are the two prior cases those of Xian and Aishu? If so, please say so.

3. Parts of the case descriptions are difficult to follow because of the several nomenclatures used. Lines 382-390 provide several illustrations of this:

• In line 382, “EP07” is used as a name for an animal. In line 389, “EP55” is described as the name of an isolate. In Table 2, “EP55” is described as a “Case”. From context, I wondered whether similar designations connected to Genbank entries. Too much work is required to decipher the information. It would be helpful to adopt a consistent scheme for such things.

• It would also be helpful to include a comprehensive expanded version of Table 2 that includes all of the animals mentioned in the paper, their birth and death years, virus and strain information, sequence accessions, lengths, and genes (for short sequences), etc.

4. A single, large “EEHV1 Loads” could be added parallel to the y-axes on the left side of Fig. 3

5. Fig. 4 would be easier to understand if it included the names of the animals, in place of, or at least in addition to codes such as “E361A_EP07”.

6. Fig. 5 would be improved by addition of animal names and where they are from (American, Zurich, or European).

7. Fig. 6 would be improved by a lighter color scheme in the upper right half and a larger font in the lower left half. The color scheme for the aligned genome fraction provides little information due to the high degree of similarity across the collection. A color gradient that ranges from 80% or 90% to 100% might be more informative.

8. The red text in Figure 7 is illegible. A larger font will increase the vertical dimension of the figure, but that is OK.

6. PLOS authors have the option to publish the peer review history of their article (what does this mean? ). If published, this will include your full peer review and any attached files.

**Do you want your identity to be public for this peer review?** For information about this choice, including consent withdrawal, please see our Privacy Policy .

Reviewer #1: No

Reviewer #2: **Yes: ** Philip E Pellett

---

## [Author Response · Author response to Decision Letter 1]

30 Oct 2024

All queries have been addressed and the responses are included in table format included with the resubmission

---

## [Decision Letter · Decision Letter 1]

10 Dec 2024

PONE-D-24-09462R1Epidemiological, Serological, and Viral Genomic Analysis of an Outbreak of Elephant Hemorrhagic Disease in SwitzerlandPLOS ONE

Dear Dr. Ackermann,

Thank you for submitting your manuscript to PLOS ONE. After careful consideration, we feel that it has merit but does not fully meet PLOS ONE’s publication criteria as it currently stands. Therefore, we invite you to submit a revised version of the manuscript that addresses the points raised during the review process. Please submit your revised manuscript by Jan 24 2025 11:59PM. If you will need more time than this to complete your revisions, please reply to this message or contact the journal office at plosone@plos.org . Please include the following items when submitting your revised manuscript:

We look forward to receiving your revised manuscript.

Kind regards,

Vishwanatha R. A. P. Reddy

Academic Editor

PLOS ONE

Additional Editor Comments:

Dear authors,

Thanks for resubmission of the reviewed manuscript, and this is now reviewed by two reviewers. After careful consideration of the reviewers comments, unfortunately, in the present form it is not acceptable in the present format, and I would like to invite you to address the comments. Please adapt the below comments.

1) The introduction needs to succinct and not a full literature review on the EEHV genome.

2) Interpretation of the data should be restricted to the discussion

3) The treatment details should be added to this paper, as these are referred to in the discussion

4) The abstract needs to clearly reflect the main findings

Best wishes,

Vishwanatha Reddy

Reviewers' comments:

Reviewer's Responses to Questions

**Comments to the Author**

1. If the authors have adequately addressed your comments raised in a previous round of review and you feel that this manuscript is now acceptable for publication, you may indicate that here to bypass the “Comments to the Author” section, enter your conflict of interest statement in the “Confidential to Editor” section, and submit your "Accept" recommendation.

Reviewer #1: (No Response)

Reviewer #2: All comments have been addressed

2. Is the manuscript technically sound, and do the data support the conclusions?

Reviewer #1: Yes

Reviewer #2: Yes

3. Has the statistical analysis been performed appropriately and rigorously? 

Reviewer #1: Yes

Reviewer #2: Yes

4. Have the authors made all data underlying the findings in their manuscript fully available?

Reviewer #1: No

Reviewer #2: Yes

5. Is the manuscript presented in an intelligible fashion and written in standard English?

Reviewer #1: No

Reviewer #2: Yes

6. Review Comments to the Author

Reviewer #1: These are some very interesting findings, but the manuscript in its present state is somewhat challenging to read. I hope my suggestions are helpful to reach a wider audience:

Introduction is 6 pages A4, please abridge to max 3 pages but referring to appropriate references. The introduction is too comprehensive, and should only cover

1) The global issue of EEHV

2) The difficulties with the genome, in broad lines

3) The presenting case, which leads into:

4) What this study is looking at

Try to remove all animal names from the text. Use Calf A, Calf B, Calf C. Bull 1, Bull 2, etc. You can have a tablet or figure in which these neutral phrases are connected to the animal’s name. By doing so, it will be easier for the reader to critically assess the course of the disease and results of what you’ve found.

Results – keep these factual by only reporting them, not interpretating them until the Discussion section, excluding the comparisons part of your study. Do not use emotive words, or interpretative words in this section.

Treatment of the calves is touched on in the discussion but details are not included. Either include the treatment in your case description, or if it is novel and warrants a separate publication, publish both papers in tandem so that you can refer too it.

Reviewer #2: Ackermann and colleagues have provided a detailed account of things learned while dealing with a slow-moving outbreak of elephant hemorrhagic disease in captive and free-living elephants. The work includes extensive use of virus genome sequencing to track what turned out to be several strains of elephant endotheliotropic herpesviruses (EEHV). This is a complex outbreak report, not a hypothesis-test. It is important that this work be archived in an internationally accessible venue such as PLOS ONE, as guidance for whoever might encounter another such situation, be it with EEHV in elephants or some other unusual combination of agent and host.

The authors are to be commended for the meticulous work done in dealing with the unfortunate clinical situations and in the depth and manner in which they have organized the compiled associated dataset and narrative, including submission of the numerous DNA sequences to Genbank.

The original reviewers made numerous suggestions for improving the manuscript. As detailed in their Rebuttal letter, the authors have done a very commendable job of responding to the suggestions.

I have nothing to add.

7. PLOS authors have the option to publish the peer review history of their article (what does this mean? ). If published, this will include your full peer review and any attached files.

**Do you want your identity to be public for this peer review?** For information about this choice, including consent withdrawal, please see our Privacy Policy .

Reviewer #1: No

Reviewer #2: **Yes: ** Phil Pellett

---

## [Editor Report · Decision Letter 2]

16 Feb 2025

Epidemiological, Serological, and Viral Genomic Analysis of an Outbreak of Elephant Hemorrhagic Disease in Switzerland

PONE-D-24-09462R2

Dear Dr. Ackermann,

We’re pleased to inform you that your manuscript has been judged scientifically suitable for publication and will be formally accepted for publication once it meets all outstanding technical requirements.

Kind regards,

Vishwanatha R. A. P. Reddy

Academic Editor

PLOS ONE

---

## [Editor Report · Acceptance letter]

PONE-D-24-09462R2

PLOS ONE

Dear Dr. Ackermann,

I'm pleased to inform you that your manuscript has been deemed suitable for publication in PLOS ONE. Congratulations! Your manuscript is now being handed over to our production team.

Kind regards,

on behalf of

Dr. Vishwanatha R. A. P. Reddy

Academic Editor

PLOS ONE